# Functional Food from Endangered Ecosystems: *Atriplex portulacoides* as a Case Study

**DOI:** 10.3390/foods9111533

**Published:** 2020-10-24

**Authors:** Lorenzo Zanella, Fabio Vianello

**Affiliations:** Department of Comparative Biomedicine and Food Science, University of Padua, Viale dell’Università 16, 35020 Legnaro, Italy; lorenzo.zanella@libero.it

**Keywords:** halophyte, *Atriplex portulacoides*, food supplement, functional food, antioxidants, sustainable food production

## Abstract

Biodiversity is a reservoir of potential sources of novel food and feed ingredients with suitable compositions for the improvement of the diet and well-being of humans and farmed animals. The halophyte *Atriplex portulacoides* occurs in habitats that are exposed to seawater inundations, and shows biochemical adaptations to saline and oxidative stresses. Its composition includes long chain lipids, sterols, phenolic compounds, glutathione and carotenoids. These organic compounds and micronutrients, such as Fe, Zn, Co and Cu, make this plant suitable as an optimal functional food that is potentially able to reduce oxidative stress and inflammatory processes in humans and animals. Indeed, many of these compounds have a protective activity in humans against cardiovascular pathologies, cancer, and degenerative processes related to aging. The analysis of its history as food and forage, which dates back thousands of years, attests that it can be safely consumed. Here, the limits of its chemical and microbiological contamination are suggested in order to comply with the European regulations. The productivity of *A. portulacoides* in natural environments, and its adaptability to non-saline soils, make it a potential crop of high economic interest.

## 1. Introduction

Brackish wetlands and salt marshes are endangered habitats of great ecological interest, and have therefore been identified as special protection areas pursuant to Council Directive 92/43/EEC, also known as the Habitats Directive. The biodiversity occurring in these environments is characterized by peculiar adaptations to tolerate soil salinity and periodic tidal flood, especially in the case of sessile organisms, such as halophytes.

These vascular plants are endowed with uncommon biochemical defenses to protect against oxidative stress, which make them potential sources of secondary metabolites of nutritional, medical and pharmaceutical interest [1,2,3,4,5]. *Atriplex portulacoides*, commonly called sea purslane, is an edible halophyte that shows these traits and is potentially suitable to be cultivated for human and animal nutrition.

The introduction of innovative crops into European agrosystems is becoming an urgent need in order to counteract the heavy loss of agro-biodiversity [6], which is related to many pivotal sustainability issues, e.g., the precipitous decline of terrestrial European insects [7]. Field-based investigations have shown that biodiversity also affects the forage quality, with important economic benefits for farmers and graziers [8]. The identification of plants with high-value biochemical composition, adaptability to climate changes, and an intrinsic potential to reconnect croplands with endangered natural ecosystems is a challenging issue. Intriguingly, *A. portulacoides* shows all of these characteristics.

The present work is aimed at the revision and discussion of the scientific literature concerning the composition and productivity of *A. portulacoides*, in order to point out its potential exploitation as functional food. Although this plant has been consumed by some populations since ancient times, it is not currently included in any official list of species intended for human consumption, and falls within novel foods according to European legislation. The present review, therefore, besides the analysis of its functional properties to favor human health, also considers historical uses and other aspects in support of its authorization as food. Consistently, issues concerning safety, toxicology and contamination limits in the case of commercialisation as food or fodder are analysed as well.

The use as forage, in addition to favoring animal welfare and promoting significant productions of this currently-uncultivated species, could allow the vehiculation of some functional compounds through food products of daily consumption, such as milk and dairy products. Importantly, *A. portulacoides* can be cultivated in saline and arid lands, and is thus an optimal plant for a sustainable agriculture.

## 2. Morphology and Systematics of *Atriplex portulacoides*

*Atriplex portulacoides* L. 1753 is a small perennial, halophile shrub, which is common along the European coasts and occurs in North Africa, Asia minor, and Western Cape province (South Africa), and has been introduced to North America [9]. Its branches are initially woody and decumbent, then soft and erected up to 20–50 cm height [10] (Figure 1), and are characterised by opposite leaves, slightly fleshy, linear-lanceolate with a full margin, and a glaucous silvery-gray colour. Its flowering period is between June and July.

*A. portulacoides* belongs to the Chenopodiacee family, whose phylogeny is complex and the object of disputed interpretations. In fact, some botanists consider Chenopodiaceae to be monophyletic because of some molecular markers [11], whereas others interpret this family as paraphyletic to Amaranthaceae, and include it therein [12].

Linnaeus described *Atriplex portulacoides* in 1753, but the position of this taxon was then revised by Moquin-Tandon [13], placing it under the genus *Obione*, and finally it was separated into the new genus *Halimione* by Aellen [14]. The genus name *Halimione*, which derives from the Greek ‘halimos’ (ἅλιμος) and means ‘seashore’ [15], is the most frequently used in literature, and is still considered valid by many authors. However, Kühn et al. [16] synonymized both *O. portulacoides* (L.) Moq. and *H. portulacoides* (L.) Aellen with *Atriplex portulacoides* L., re-establishing the original nomenclatorial act by Linnaeus. Even more recently, Kadereit et al. [17] resurrected the genus *Halimione* because of some molecular and morphological markers that are distinctive from *Atriplex*, which is the largest genus of the Chenopodiaceae family, with about 300 species. In this controversial background, *Atriplex portulacoides* is acknowledged as a senior synonym of *Halimione* in the International Plant Names Index, and this taxon name will be therefore adopted in the present work.

Interestingly, the species name *portulacoides* is obtained by combining ‘Portulaca’, a genus name of a species belonging to the family Portulacaceae, with ‘eidos’ (εἶδος), a Greek word that means ‘appearance’ [15]. The species name, therefore, means ‘similar to Portulaca’, recalling the leaf similarity with the common purslane (*Portulaca oleracea* L.), which belongs to another family of the order Caryophyllales. Importantly, this morphological resemblance and nomenclatorial assonance have induced some non-specialists (for instance, in patent applications) to include *A. portulacoides* between the plants of the so-called ‘purslane family’ (see Bair et al. [18] for a correct interpretation of this group), but this association in erroneous, and is a potential source of relevant mistakes.

## 3. Use History of *A. portulacoides*

*A. portulacoides* can be eaten both raw and cooked [19]. The dietary use of halophytes by coastal populations dates to the late Neolithic period, as evidenced by the discovery of *A. portulacoides* amongst ancient charred remains of food in northern Holland [20]. These findings show that this halophyte was habitually consumed, and perhaps cultivated, around 2500 cal BC.

According to Custódio et al. [21], *A. portulacoides* is collected from the wild by professional foragers, and is sold in specialised online platforms for local restaurants and gourmet cuisine. In Italy, this herb was traditionally eaten in salads, or boiled. In the Marche and Sardinia regions, it entered into some recipes based on fish [22]. Its buds can be preserved in vinegar, as occurs for capers [10]. An ancient Sardinian recipe adopted by fishermen inhabiting the lagoon area of Cabras, locally called ‘*mreca*’, is based on mullet (*Mugil cephalus*) boiled in brackish water and then wrapped with fronds of *A. portulacoides* to make a sort of ‘cocoon’. This preparation shows preservative properties for a few days, and was traditionally used as a ready-to-eat fish during fishery trips [23].

## 4. Eco-Physiology of *A. portulacoides* and the Implications for Its Chemical Composition

The knowledge of the ecophysiology of this plant is important not only to explain its high content of biologically active compounds, but also to evaluate its cultivability for food purposes.

*A. portulacoides* is adapted to live in soils with varying degrees of salinity, and can survive short periods of submersion in saltwater [24,25]. It can occupy salt marsh and coastal habitats where other species cannot survive or are competitively less efficient. Crain et al. [26] showed that many halophilous plants do not necessarily require saline soil to live, but they are able to adapt to saline environments and prevail over most potential competitors. This is the case for *A. portulacoides*, which finds optimal conditions in moderately saline environments but can successfully develop in non-saline soils, as inferred from several studies. Under controlled conditions, the net organic matter production (ash-free dry weight) of plants cultivated in the absence of salt showed a productivity not far from the best performances of others treated with 85–170 mM NaCl (about 5 to 10 PSU (Practical Salinity Unit)), whereas it decreased exceeding 200 mM NaCl (about 12 PSU), with a relevant impact from 500 mM NaCl (about 29 PSU) [25]. In another experiment, groups of plants were treated for 2 weeks with different salt concentrations (0–15 PSU), and the best growth was recorded at 5 PSU [19]. Finally, Redondo-Gómez et al. [27], recorded the maximum relative growth rate (RGR) at 200 mM NaCl.

In order to overcome the negative osmotic pressure due to the salty environment, halophytes accumulate various metabolites, including quaternary ammonium compounds; ternary sulfonium compounds; proline; and soluble carbohydrates, such as sucrose, sorbitol, mannitol, and pinitol, etc. [28]. The production of osmolytes varies among different halophytes, regarding both the chemical species and relative concentrations. Glycine-betaine (a quaternary ammonium compound, also called simply ‘betaine’) and proline are organic osmolytes that are dominant in Chenopodiacee and Graminee [29]. The betaine concentration can reach 238 µmol/g dry weight (DW) in epigean biomass, corresponding to 27.9 mg/g DW, and about an 85:1 molar ratio with respect to proline, according to Storey et al. [29]. Betaine is an electrically neutral compound at pH 7.0 (zwitterion), is very soluble, and is suitable to form hydrogen bonds with a high number of water molecules (solvation sphere). For this reason, it is very effective in retaining tissue water, and an excellent moisturizing ingredient for cosmetic use [30].

*A. portulacoides* also reacts to mild saline stress (10 PSU) by increasing its content of proline and antioxidants within a few hours, then decreasing to low values in the subsequent days [19].

In addition to organic osmolytes, salts passively introduced with water contribute to enhance the tissue’s osmotic pressure. Some plants developed in soils with π = 450 mOsm presented a lymph with a calculated osmotic pressure of π = 1087 mOsm, to which K contributed 72 meq/l, Na contributed 532 meq/L, Cl contributed 433 meq/L, and Ca and Mg were not detectable [29]. Redondo-Gómez et al. [27] found that NaCl is compartmentalised according to the following concentration gradient: leaves > stems > roots. In the leaves, Na and Cl accumulate at increasing concentrations with aging, whereas Mg and Ca show the opposite trend [24,29]. Plants treated with increasing salinities for 40 days showed that Na and Cl increased mainly in the leaves, whereas K and Ca decreased, the latter to a lesser extent [31]. Plant composition and saline content can vary significantly in relation to the season, the geographical locations and the morphology of the development environment.

Some adaptations to salinity have similarities with those aimed at resisting drought, as—in both cases—the plant needs to absorb and retain water under the conditions of an adverse gradient. Therefore, it is not surprising that *A. portulacoides* tolerates temporary water shortage, despite the fact that it develops in salt marshes and other coastal humid environments. Indeed, its zonation in the typical environment reflects its preference for well drained soils and limited exposure to prolonged flooding [24,32]. This trait is consistent with the absence of aerenchyma in its roots, which presupposes a well oxygenated soil, unlike other halophytes which, instead, occupy the lower salt marsh and contribute to the substrate aeration, thanks to their radical aerenchyma [32].

### Metabolism of Heavy Metals

An adaptive trait related with the accumulation of mineral osmolytes is the capacity of *A. portulacoides* to concentrate seawater metal cations, which is of interest for some micronutritional applications. However, in addition to naturally occurring metals, halophytes are currently exposed to toxic metals derived from anthropogenic pollution [33]. Therefore, attention must be paid to the site from which plants intended for food use are harvested. Several studies have shown that the high levels of heavy metals present in some European estuarine and lagoon environments have an impact on the composition of *A. portulacoides,* and on its lipid profile. For instance, linolenic acid content decreased in favour of palmitic acid and a reduction of mono- and di-galactosyldiacylglycerol was observed as effects of chronic exposure to environmental pollutants, mainly heavy metals with oxidizing activity [34]. Interestingly, *A. portulacoides* can sequester and/or compartmentalise metals, binding most of them to the structural components of the cell wall (pectins, lignin, cellulose, other polysaccharides), or to special proteins [33,35,36,37] according to the following affinity scale: Zn > Pb > Cu > Ni > Co > Cd [35]. Due to this trait, the use of halophytes for the phytoremediation of heavily polluted wet areas has been proposed [35,38]. A case study of the accumulation/repartition of metals in the different parts of the plant and cell compartments is shown in Figure 2.

Phytochelatins are amongst the inducible metabolites that are suitable for metal binding. These compounds are peptides that are rich in cysteine, with general structure of (γ-Glu-Cys)_n_-Gly, in which ‘n’ usually varies between 2 and 5 [39]. Phytochelatins can bind ions by means of coordination bonds with the thiol functional groups, as occurs with other abundant mono-thiol compounds, i.e., glutathione (GSH), γ-glutamyl-cysteine and cysteine molecules [39]. These functional groups are also extremely active in the broad-spectrum prevention of damage from oxidative stress.

## 5. Chemical Biomass Composition and Related Nutritional Value

The leaves are the most interesting plant compartment for nutritional purposes and—despite the fact that they are only 2.3% of the total dry biomass due to their high content in water— they show the highest content of nitrogen, which is indicative of their protein content (Table 1).

*A. portulacoides* has been extensively studied with regard to ecology and physiology, but insufficient data are available concerning its quantitative and qualitative protein content. Some crude protein data, inferred from their total N content, and other approximate composition values are shown in Table 1, from which it can be deduced that the leaves’ crude proteins range from 5 to 11% DW, with a modal value around 10%, whereas the lipids vary from 1.7 to 3.2% DW, depending on the plant compartment being considered. Boughalleb et al. [45] quantified the lipids that are extractable with different solvents, obtaining yields on DW biomass of 1.8% with petroleum ether, 2.1% with acetone, and 3.1% with methanol. An indicative crude lipid content of about 2% DW is therefore plausible as a general case.

Fresh epigeal biomass generally comprises > 80% moisture, with an average value around 87.5%, whereas ashes increase with environmental salinity, ranging from 27 to 49% in the shoots’ dry biomass [25]. The high ash content is directly related to the mineral concentration of the tissue, which in turn depends on what the plant needs to counteract the environmental osmotic pressure.

### 5.1. Lipids

The lipid fraction of the biomass is of special interest for human and animal nutrition, as it comprises many active compounds, such as long chain polyunsaturated fatty acids (PUFAs), carotenoids, and sterols, etc.

An important contribution in this regard was disclosed by Vilela et al. [44], who worked on samples from estuarine environments of Portugal (Table 2). The roots showed the highest lipid content (3.2% DW), followed by the leaves (2.1% DW) and the stems (1.7% DW) [44]. The main lipids detected in the leaves were saturated fatty acids (0.062% DW), whereas unsaturated fatty acids were 0.018% DW, and were almost totally composed of oleic acid (0.012% DW) and linoleic acid (0.006% DW). However, Maciel et al. [46] obtained a leaf fatty acid profile composed of 60% PUFAs, of which 43.5% were due to linolenic acid and 14% linoleic acid, and a n-6/n-3 fatty acids ratio equal to 0.32, which is particularly favorable for nutrition purposes. Small fractions of uncommon long chain saturated fatty acids, containing up to 24 carbon atoms, were also present. The same study showed the appreciable presence of phospholipids and glycolipids of high biological value.

It seems that the ratio between the unsaturated and saturated fatty acids is affected by environmental contaminants, especially sedimentary heavy metals, and other generators of reactive oxygen species. Thus, this ratio has been proposed as a biomarker of pollution [34]. Amongst the compounds of interest shown in Table 2, long chain aliphatic alcohols (LCAA) appeared, which are generally constituents of epicuticular waxes. In leaves, about 46% of these were represented by 1-octacosanol (C_28_H_58_O), 24% by 1-hexacosanol (C_26_H_54_O), and 22% by 1-triacontanol (C_30_H_62_O).

Amongst sterols, β-sitosterol and its hydrogenated derivative, β-sitostanol, prevailed; both are known as anti-cholesterolemic agents. The roots showed a higher content of sterols/triterpenoids than epigeal compartments, which can amount to about 3.2 g/kg DW, of which about 87% was hop-17(21)-en-3-one, as a sum of both the ketone and enolic form.

Amongst Chenopodiaceae, *A. portulacoides* stands out for its high content of ecdysone-related compounds, which are not degraded by cooking and can produce an anabolic effect in vertebrates [47,48]. The prevalence of 20-hydroxyecdisone in the epigean parts was found, whereas its derivative 20-hydroxyecdisone-septanoside was in roots [49]. The adaptive meaning of ecdysteroids is related to their interfering action on the metamorphosis and moulting of potentially harmful phytophagous arthropods [47].

### 5.2. Carotenoids

Carotenoids are accessory pigments that enhance the absorbance of radiant energy in the chloroplast, but also protect the structural integrity of thylacoids from oxidative damage [50]. They can be ingested by humans alongside their normal diet for health improvements [51].

Anjum et al. [52] quantified the carotenoids of *A. portulacoides* leaves in 5 µg/g fresh weigh (FW), corresponding to 40 µg/g DW if a moisture of 87.5% is assumed. The same study showed an increase of carotenoids in some plants grown in highly contaminated soils, in response to mercury toxicity, up to more than twice than that of the controls. Higher contents of carotenoids (around 50 µg/g FW) were detected by Duarte et al. [53] in plants maintained under favourable conditions, which decreased drastically following the exposure to saline stress. This was partially counterbalanced by an increase in anthocyanins and beta-cyanins.

Carotenoids from 30 to 100 µg/g FW (approximate values estimated from graphical representations) were reported by Duarte et al. [54] in plants stressed by frequent tidal floods, which decreased with the reduction of the flood frequency. In another study, Duarte et al. [55] quantified the leaf carotenoids in about 109–230 µg/g FW, indicatively corresponding to 0.87–1.84 mg/g DW, assuming a moisture of 87.5%, amongst which zeaxanthin was the most abundant. Intense changes of temperature produced relevant variations of some carotenoids, with different effects depending on the compound and the treatment considered [55]. The most abundant carotenoids were β-carotene, auroxanthin, antheraxanthin, lutein, violaxanthin, and zeaxanthin [53,55].

### 5.3. Phenolic Compounds

*A. portulacoides* synthesizes numerous phenolic compounds, and in higher amounts if exposed to environmental stresses [56]. According to data published by Vilela et al. [44], total phenols were more abundant in root extracts (142.1 mgGAE, (gallic acid equivalent)/g dry extract) than in extracts from the stems and leaves (52.1 and 48.3 mgGAE/g dry extract, respectively), but the leaves showed the highest amount of phenolic compounds per DW of extracted biomass (19.9 mgGAE/g DW). Sulphonated and/or glucuronated derivatives of isorhamnetin were the main compounds detected (Table 2). Phenolic compounds conjugated with sulphate groups were about 3.0 g/kg DW in leaves, which is an extremely high value when compared to other plant sources [44].

Ethyl acetate, methanol and water extracts of epigeal parts of *A. portulacoides* were studied by Zengoin et al. [57]. The authors recorded a total phenolic and total flavonoid content in ethyl acetate extract of 14.59 ± 0.21 mgGAE/g of extract and 6.19 ± 0.06 mgGAE/g of extract, respectively. Furthermore, this study disclosed the presence in the methanol extract of an amount of quinic acid that was even higher than rhamnetin, followed by other minor compounds, such as p-coumaric acid, 4-OH benzoic acid, salicylic acid, malic acid, tr-caffeic acid, vanillin, hesperidin, rutin, rosmarinic acid, chlorogenic acid, and gallic acid [57].

## 6. Productivity and Cultivability

The availability of wild plants is generally limited, especially if they develop in environments that are at risk of disappearance. Thus, their potential economic exploitation depends on the option of a profitable cultivation. Some field studies in natural environments showed that the primary productivity of *A. portulacoides* could be of interest for agricultural exploitation. Jensen [25] studied the ash-free DW relative growth rate (RGR) in experimental hydroponics under different salinities, from seedlings until the second pair of leaves had emerged. In a 9 week treatment, the RGR ranged between 43 and 53 mg/g/day, with a salinity between 0 and 515 mM NaCl (about 30 PSU), while it decreased to 27 mg/g/day at 690 mM NaCl (about 40 PSU). Under different experimental conditions, Redondo-Gómez et al. [27] detected an ash-free RGR between 20 mg/g/day (control in the absence of salt) and 40 mg/g/day at 200 mM NaCl (about 12 PSU), which dropped to about 5 mg/g/day at 700 mM NaCl (about 41 PSU) (values estimated from graphic representations).

As a whole, although they were recorded under different experimental conditions, these data are not far from the RGR performances of some alfalfa cultivars, which showed RGR between 20 and 30 mg/g/day [58]. Importantly, the RGR of *A. portulacoides* was also high in the absence of NaCl, thus resulting in its being adaptable to conventional agricultural lands.

Some biomass density and productivity data were estimated in the wild using different methods, amongst which the following were obtained with the Smalley method, which is the most frequently adopted method. Data concerning the epigean biomass can range between 598 g/m^2^ DW [42] and 1363 g/m^2^ DW [59], whereas (in Portugal) the values of RGR were estimated to be more favourable in winter–spring, with 7.9 mg/g/day, or 0.8% per day [42]. The productivity data were assessed in the salt marshes of some European countries: 952 g/m^2^/year DW was found in the Cantabrian Sea (Biscay Gulf, Spain) [60]; 790–1434 g/m^2^/year DW was found in a Danish salt marsh (Krabbendijke) [61].

Overall, the above reported data suggest that, under natural conditions, an approximate primary productivity of 1000 g/m^2^/year, or 10 t/ha/year DW, might be calculated. This estimation is consistent with the most conservative data reported for other halophyte cultivations investigated within projects aimed at the agricultural development of arid or saline regions. Sardo [62] reported on productivity values of 24 t/ha DW for a generic *Atriplex* sp. Le Houérou [63] reported on halophyte forage productions of about 20 t/ha/year DW in irrigated saline environments, which dropped to 2–10 t/ha/year DW in semi-arid environments (rainfalls of 400–800 mm/year). Yields of 18 t/ha/year DW were reported for *Atriplex lentiformis* grown at a salinity of 500 mM, and 21.3 t/ha/year DW were reported for *Atriplex triangularis* at salinity of 150 mM [64].

### 6.1. Exploitation as Forage

The extensive grazing of livestock, mainly sheep, on extensive areas vegetated by halophytes has been practiced since ancient times in some European salt marshes. In the Wadden Sea (the coast of the North Sea), natural salt marsh has been exploited as grazing lands since 2600 years ago [65]. In Mont Saint-Michel bay (France), three-quarters of the salt marsh area (30–32 km^2^) were grazed by 17,000 sheep in recent times [66]. In natural wetlands, not only sheep and cows graze halophytes comprising *A. portulacoides*, but also wild species, such as wild boars [67], geese and hares [68,69].

These investigations attest that halophytes can be used as pasture, but were focused on the ecological impact of grazing, whereas nothing was reported about the quality of the breeding products. The latter is expected to be increased due to the active compounds comprised in the *A. portulacoides* biomass. Some nutritional criticalities can arise for the high content of ash and salts, the low metabolisable energy, and the possible presence of undesired secondary metabolites [70]. Nonetheless, the use of halophytes for animal nutrition has long been considered an important opportunity for economic development [3], especially for arid geographical areas or in the need to irrigate using salty water [62,63,71,72]. The nutritional value as forage can be inferred by comparing some data of a hypothetical production of *A. portulacoides* with those of alfalfa and other fodders (Table 3). Assuming a conservative yield estimation of 10 t/ha DW, based on the data discussed above, *A. portulacoides* shows productive parameters close to alfalfa, which is the most productive grassland crop in terms of dry matter and proteins [73]. The protein content is obviously lower than the comparative legume, which shows the highest value of this parameter, i.e., two times higher in soy and three times in pea sown in the fall [73]. However, the protein intake provided by *A. portulacoides* is higher than the minimum required for weight maintenance by sheep or cows, i.e., 7–9% [3], and is in line with the highest content of alternative forages considered in Table 3. Importantly, the protein content is lower than the minimum required for lactating animals, equal to 14–18% [3].

The lipid content of *A. portulacoides* is higher or comparable to that of fodder alternatives (Table 3), but with a more interesting quality profile (Section 5).

Chenopodiacee are a good source of protein, sulfur and minerals, while providing insufficient metabolisable energy [3]. The nitrogen deriving from betaine, which assumes relevance in *A. portulacoides*, is generally metabolised up to 50% by the rumen microbiome, and this fraction can increase in diets with low metabolisable energy, whereas proline can be easily absorbed at the ileum level [3]. The high ash content is mainly due to salts and metal ions, which contribute to the micronutritional value, and make *A. portulacoides* an optimal ingredient for the improvement of the nutritional profile of mixed forages [76]. In Table 4, some essential minerals required by sheep and cattle are compared with the composition of *A. portulacoides* and some other plants that are used to feed animal husbandry. Whenever possible, data on *A. portulacoides* were chosen from those referring to plants grown in the absence of salt, since they are considered more representative of potential cultivated crops. As is consistent with its high content of ashes and its metal storage capacity, *A. portulacoides* is rich in some elements of which other forages are deficient with respect to animal needs. Therefore, if used in proper proportions in the formulation of forage blends, this halophyte can be a useful dietary supplement for livestock nutrition. In particular, the content in S can support the requirement of sheep reared for wool production. Among metals, *A. portulacoides* is a valid source of Fe, Zn, Co and Cu.

### 6.2. Human Nutrition

*A. portulacoides*, although it is considered an almost-forgotten traditional food, still appears in culinary cookbooks dedicated to alimurgical plants, in some gourmet preparations, and among the proposals of chefs dedicated to creative reinterpretations of traditional cuisine. Its biochemical composition comprises many molecules that are useful for the promotion of human health, although important information is still missing concerning, for instance, the composition and digestibility of its plant proteins.

The lipid fraction is dominated by oleic and linolenic acid [44,46], both of which have antioxidant, antinflammatory and anti-trombotic activities [81,82]. Among the lipophilic active molecules, attention should be paid to LCAAs, sterols and ecdysone-derived steroids. Indeed, 1-octacosanol showed beneficial activities, such as antiradical activity, protection from parkinsonism by regulating the proNGF (precursor of nerve growth factor) and NGF (nerve growth factor) signals [83], the stimulation of the activity of creatine phosphokinase in plasma and citrate synthetase in the muscle of rats subjected to stress [84], and the reduction of plasma triglycerides [85]. Another LCAA, 1-triacontanol (C_30_H_62_O), showed anti-inflammatory properties following topical skin applications [86]. Sterols and stanols, β-sitosterol and β-sitostanol, exert an anti-cholesterolemic action due to their low intestinal absorption, combined with a cholesterol sequestering ability, which increases fecal cholesterol and bile acid excretion [87]. According to the European Food Safety Authority (EFSA) a consumption of 1.5–2.4 g/day of phytosterols and/or stanols is recommended in the order to reduce blood cholesterol [88]. Long-term feeding tests showed that β-sitostanol is more active than β-sitosterol [89], but the latter is by far the most abundant phytosterol in the human diet, and is credited with numerous beneficial effects, although not all have been sufficiently demonstrated. Amongst the most relevant are: cardiovascular protection, immune system modulation, the prevention of some types of cancer and rheumatoid arthritis, and the prevention of hair loss [88]. Hopenone is an uncommon triterpenoid ketone with an anti-proliferative action, which is deemed potentially useful for cancer prevention [90,91]. Amongst steroids, ecdysone septanoside-derivatives showed antioxidant activity, the inhibition of cholinesterase (potentially functional for preventing hypertension and related heart disease) and a mild inhibiting activity against some bacteria strains of interest [49].

The useful compounds for human health also include carotenoids, which reduce the incidence of cardiovascular diseases and the risk of cancer in some organs: breast, lung, various compartments of the female genital system, the prostate and the colon [92]. Amongst the most represented xanthophylls are lutein, zeaxanthin, violaxanthin and antheroxanthin [53], all of which are strong antioxidants, the first two of which are active in the protection of the eye retina [93]. The polyphenol content is quite relevant, and is characterized by uncommon sulphated flavonoids, mainly isoramnetin derivatives, which showed antithrombotic activities [44], colon–rectal anti-carcinogenic properties [94], anti-inflammatory activity by COX2 (cyclooxygenase-2) inhibition, the activation of the Nrf2 factor (NF-E2 p45-related factor 2) [95], and the inhibition of the inflammatory signalling mediated by NF-kB (nuclear factor kappa-light-chain-enhancer of activated B cells) and MAPKs (mitogen-activated protein kinases) [96].

Other compounds of interest are glycolipids, which are also in sulphate form, e.g., mono- and di-galactosyldiacylglycerol, sulfoquinovosyldiacylglycerol and hexosylceramide [46], which disclose a significant anti-inflammatory activity [97,98].

Amongst the hydrophilic antioxidants are glutathione, other monothiol compounds and phytochelatins, which in turn could be decomposed to single- or oligo-peptides by digestion. GSH, and probably also γ-glutamyl-cysteine, can be absorbed through the gastrointestinal tract by Na-dependent and Na-independent carriers, especially at the level of the small intestine, despite some doubts which remain on the absorption of GSH [99]. However, GSH absorption through the oral mucosa was shown using oral mucosa models, while plasma GSH levels increased significantly in a clinical trial conducted with rapid/slow release GSH preparations [100].

The antioxidant phytocomplex of *A. portulacoides* is consistent with its anti-inflammatory properties, which it is one of the strengths of this food. Extracts of this plant have been shown to reduce the nitric oxide release by human macrophages stimulated with sulphate lipopolysaccharides [101]. Besides this, topical treatments with hydroalcoholic and aqueous extracts of this plant significantly reduced the release of interleukin IL-1α in human skin samples stimulated with 2% sodium dodecyl sulphate [102]. In vitro tests with shoot extracts showed an inhibitory activity on some enzymes, such as butyrylcholinesterase, tyrosinase, amylase and glucosidase [57], with potential effects on the production of acetylcholine, melanin, and on the glycaemic control, respectively.

All of these properties are of interest for human nutrition, especially because this plant can be eaten raw, without undesired compositional modifications due to cooking processes. Furthermore, *A. portulacoides* is a source of some important minerals for micro-nutritional purposes, namely Mg, K, Ca, Fe, Zn, Cu and Mn. The bioavailability of these elements for humans has not been assessed yet, and metals bound to phytochelatins and other peptides could be released following protein digestion. Table 5 shows the mineral intake levels recommended for the adult Italian population, while the last line shows the related intake provided by 100 g FW *A. portulacoides*. It should be noted that this modest ration takes at least 10% of the recommended intake of most minerals, exceeding 40% in some cases, e.g., Fe and Mg.

## 7. Safety Specifications and Recommendations

*A. portulacoides* is a perennial plant that has edible parts, mostly represented by the leaves and tender stems, which are assimilable to many other vegetables for safety concerns. However, its prerogative to absorb and concentrate some substances, especially metals, makes it potentially dangerous if collected from contaminated environments. Microbiological contaminations should also be considered, as this plant is suitable to be eaten raw. Table 6 shows some chemical and microbiological limits provided by the current legislation for some foods that are comparable to *A. portulacoides*. Microbiological analyses performed by Mangia et al. [23] on *A. portulacoides* that was freshly collected and then used for the preparation of a traditional *mreca* (Section 3) showed quite high microbiological loads, which were possibly related to the soil microbiome and/or inadequate handling, confirming the importance of carefully assessing the site of cultivation and the need to implement adequate hygiene conditions throughout the product’s processing.

The nitrate content also deserves a special comment due to the relevance it assumes in some leafy vegetables. Some studies have shown that *A. portulacoides* probably transforms nitrates into organic osmolytes, such as betaine [25,29], and thus their concentration in the leaves varied inversely to salinity. This peculiar metabolic trait should also prevent problems concerning high nitrate levels under conditions of fertilised cultivation. In any case, the nitrate concentration must respect the limit of 2500–3000 mg/kg FW prescribed for spinach and lettuce in summer crops by the Commission Regulation (EC) No. 1881/2006, and the subsequent amendments of the Regulation (EC) No. 629/2008.

Betaine is the main organic osmolyte in *A. portulacoides,* and the possible end metabolite of nitrate exceeding protein synthesis requirements. Rats treated with high betaine doses (426 to 2442 mg/kg body weight (BW) per day), for 12 months, resulted in the lowest mean corpuscular haemoglobin and mean corpuscular volume, whereas their blood platelets were increased [103]. Obese human subjects with metabolic syndromes treated with 4 g/day betaine showed an increase of blood total cholesterol and LDL (low density lipoprotein) levels; however, the same dosage did not produce any adverse effect in healthy subjects, nor at intakes of 3 g/day [103]. Therefore, considering 4 g/day betaine as a reference point, and applying a cautelative 10 factor decrement, an amount of 400 mg/day betaine (i.e., about 6 mg/kg BW per day for adults), in addition to the background exposure (830 mg/day for the adult European population), was considered to be safe by EFSA [103]. Interestingly, the choline intake must also be taken into account, since over 50% of this compound is irreversibly oxidized to betaine in the human body.

Storey et al. [29] estimated a betaine content of 27.9 mg/g DW in *A. portulacoides* which was developed in a natural saline environment, whereas values of about 1.7 mg/g FW are reported in the absence of salt, according to graphic results by Benzarti et al. [31]. Consuming a ration of 100 g FW (assuming 87.5% moisture), the betaine intake would vary from 349 mg/100 g FW to 170 mg/100 g FW, respectively, i.e., values that are by far within the limits recommended by the EFSA.

## 8. Toxicology

The edibility of *A. portulacoides* is supported by its use history as a traditional food, although it is limited to some coastal populations and is currently almost abandoned, and as pasture for sheep and cattle (Section 6.1). Furthermore, cytotoxicity tests carried out for 72 h with aqueous, methanolic and chloroform extracts of *A. portulacoides* by Rodrigues et al. [101] on hepatocellular carcinoma HepG2 cell cultures, showed the total absence of toxicity. The same study also comprised highly lipophilic extracts (ether or hexane), which reduced the cell survival to 20–30% following the strongest treatment, i.e., 125 µg/mL. A cell survival of < 70% is required for the consideration of a compound as potentially toxic [104], and this did not occur despite the protracted treatment period (72 h).

Some rough transpositions are proposed here, which are aimed at the estimation of the food intake necessary to reproduce, in vivo, an equivalent treatment of highly lipophilic compounds. If we express the treatment in terms of the biomass from which the highly lipophilic extracts are obtained (assuming a yield < 5% of DW biomass, according to the personal experience of one of the authors), 125 µg/mL might correspond to a dose that is not lower than 2.5 mg/mL as DW biomass, or 20 mg/mL as FW biomass (calculated on 87.5% moisture). In order to expose the human body to an equivalent treatment, an intake of *A. portulacoides* FW biomass of at least 20 g/kg BW might be estimated, i.e., a value that is more than tenfold the suggested 100 g food ration. In light of these results, it is reasonable to state that there is no evidence of toxicity for *A. portulacoides*.

## 9. Conclusions and Perspectives

Modern nutrition science is always looking for novel ingredients and foods that are suitable to improve human (and animal) diets, not only in order to meet the energy and essential nutrient needs, but also to preserve and enhance the state of our health. The latter issue is related with the intake of biologically active compounds for the prevention or delay of cell damage due to environmental stresses or the physiological aging processes. In this context, the research is focused on foods that are rich in vitamins, antioxidants, immunostimulants, anti-inflammatory substances, and natural regulators of blood glucose and lipemic metabolism, etc. Moreover, in accordance with the principles of sustainable development and the green economy, food sources should respect the natural environment, and should not contribute to climate change. Halophytes, as with many organisms that have adapted to stressing environments, have developed ecophysiological adaptations based on antioxidant bioactives [4], and on the ability to concentrate some minerals of nutritional interest. *A. portulacoides* shows the properties typical of halophytes, but—in addition—it can grow in non-saline soils, and is therefore suitable to be cultivated in conventional agricultural lands. In the latter case, this plant seems to have the potentiality to develop under poor irrigation, allowing the exploitation of arid and even marginal saline lands. Thus, in view of the ongoing climate change, its production could offer an interesting opportunity for agricultural diversification, both for production aimed at the human diet, and to be used in properly balanced forages.

Concerning the composition of *A. portulacoides*, particular attention should be paid to its content in PUFAs, sterols and phenolic compounds. These active molecules appear to be suitable for being assimilated by dairy animals, and—at least partly—are transferred into milk, fortifying this food which is widely consumed on a daily basis. The impact on human nutrition deriving from the use of this fodder could, therefore, be even greater than the direct consumption of the plant carried out on an occasional basis. The improvement of the lipid profile of milk obtained by integrating bovine feed with non-traditional ingredients has already been demonstrated. For instance, the decrease of the n-6/n-3 fatty acid ratio of milk and ripened cheese was successfully obtained by supplementing the cows’ diet with extruded flaxseed [105].

The present review highlights that *A. portulacoides* possesses the characteristics to become a functional food which is useful for improving human well-being and for the diversification of agricultural productions in a sustainable way. This justifies further investments in research, which are necessary because several nutritional and functional issues of *A. portulacoides* have not been investigated yet, including its protein quality and composition, vitamin content, fruit composition, the bioavailability of the compounds of interest, its overall digestibility, and its effect on mammalian metabolism and serum parameters, etc. Many results that were obtained from in vitro experimental models need to be confirmed in in vivo tests, especially for phytocomplexes that are rich in compounds, whose metabolic responses depend on the sum of their activities. For instance, extracts tested on ex vivo skin samples produced an increase of melanogenesis [102], in spite of the inhibiting activity on tyrosinase shown in vitro [57].

Furthermore, cultivation trials and the assessment of the related yields have not been carried out yet. As this plant is perennial, the harvest should therefore be performed by taking only the upper part of the canopy, preserving the roots and most woody part of the stems in a vital state for new shoot production. The actual productivity, seasonality and composition of the harvested biomass, which would include the fruit—if the harvest is carried out in the autumn—are still unknown.

## Figures and Tables

**Figure 1 foods-09-01533-f001:**
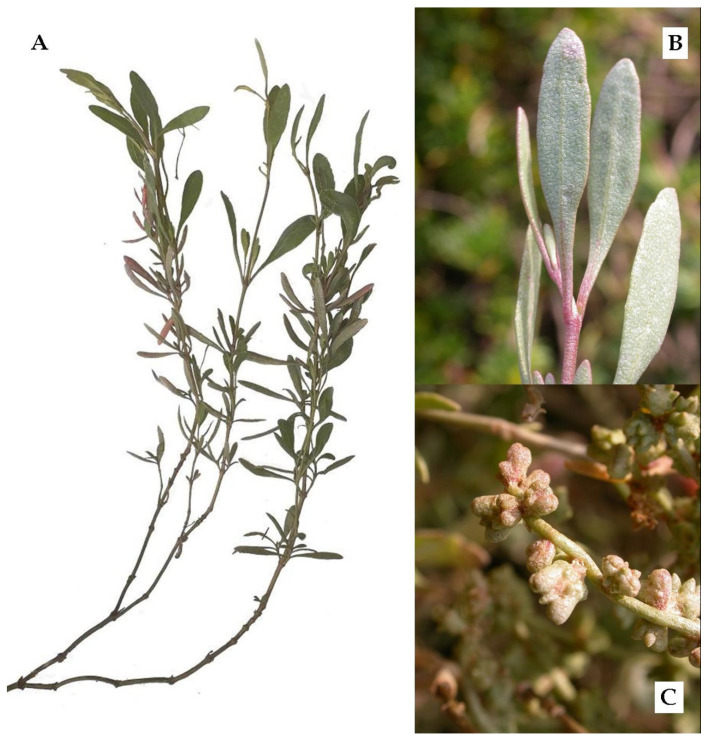
Morphology of *A. portulacoides*: (**A**) branching and decumbent habit; (**B**) leaf close up; (**C**) freshly formed fruits in autumn.

**Figure 2 foods-09-01533-f002:**
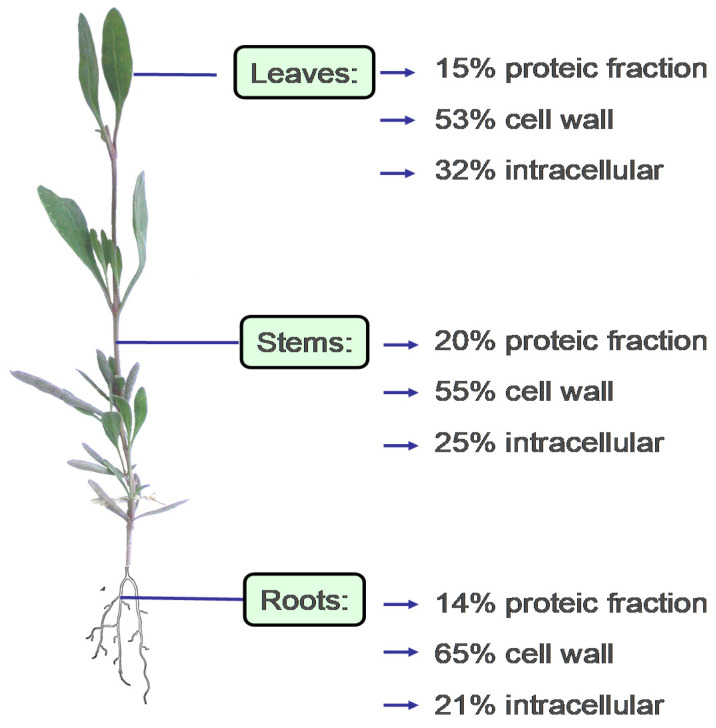
Metal distribution (leaves, stems, and roots) and compartmentation (cell wall, proteic fraction, and intracellular location) in *A. portulacoides* (redrawn and modified from Sousa et al. [35]).

**Table 1 foods-09-01533-t001:** Dry biomass and approximate composition values of *A. portulacoides* biomass (FW = fresh weight, DW = dry weight).

	Leaves	Stems	Shoots (Leaves + Stems)	Roots	Reference
Biomass (g/m^2^)	212.8	1345.6		7686.1	[40]
Nitrogen (% DW)	1.63	1.23		0.75	[40]
Carbon (% DW)	30.7	40.0		15.0	[40]
Moisture (% FW)	87.5–90.9 *				[31]
			85.9–86.8 *	61.5–80.3 *	[41]
			85.9		[29]
Protein (% DW)	5.6–9.4 **	3.1–4.4 **		6.0–7.5 **	[42]
	10.2% **	7.7 **		4.7 **	[40]
	10–11.2	6.6		3.5	[33,43]
Lipid (% DW)	2.1	1.7		3.2	[44]
Ash (% DW)	32–40 *	12–15 *		12–24 *	[27]
			30–35 *	10–12 *	[41]
			27–49	9–42	[25]

* approximate value extrapolated from graph. Data from [41] selected for plants treated with 0–25% seawater; ** the reference provided only the N content, from which the protein value was estimated here using a 6.25 nitrogen-to-protein conversion factor.

**Table 2 foods-09-01533-t002:** Main compounds (mg/kg DW) identified in extracts from the leaves, stems and roots of *A. portulacoides* (from Vilela et al. [44], modified).

Lipidic Compounds (mg/kg of Dry Material)	Leaves	Stems	Roots
Fatty acids	798	252	374
Saturated	619	193	316
Hexadecanoic acid	99	81	75
Hexacosanoic acid	66	13	18
Octacosanoic acid (montanoic ac.)	230	32	70
Triacontanoic acid (melissic ac.)	147	20	90
Other saturated fatty acids	77	47	63
Unsaturated	179	59	58
Octadec-9-enoic acid (oleic ac.)	115	32	38
Octadeca-9,12-dienoic acid (linoleic ac.)	64	27	20
Long chain aliphatic alcohols	546	82	274
1-Hexacosanol	132	18	53
1-Octacosanol	252	33	98
1-Triacontanol	122	8	31
Other long chain aliphatic alcohols	40	23	92
Long chain n-alkanes	106	9	15
Sterols/triterpenoids	323	609	3245
Cholesterol	66	5	19
Campesterol	37	4	9
Stigmasterol	20	169	n.d.
β-Sitosterol + β-Sitostanol	148	n.d.	291
Spinasterol + β-Sitosterol + β-Sitostanol	n.d.	122	n.d.
Schottenol	52	30	108
Hop-17(21)-en-3-one	n.d.	169	2102
Hop-17(21)-en-3-one (enolic form)	n.d.	110	716
Other identified and unidentified compounds	273	130	243
**Total detected compounds**	**2066**	**1084**	**4151**
**Phenolic Content (mg/kg of Dry Material)**	**Leaves**	**Stems**	**Roots**
Isorhamnetin-glucosyl-rhamnosyl-glucuronide isomer	457	–	–
Isorhamnetin-sulphate-pentoside isomer	67	–	–
Isorhamnetin-sulphate-glucosyl-rhamnoside	1190	–	–
Diosmetin-sulphate-glucuronide	422	19	–
Isorhamnetin-sulphate-pentoside isomer	384	18	–
Isorhamnetin-glucosyl-rhamnoside isomer	107	38	–
Isorhamnetin-sulphate-glucosyl-rhamnosyl-glucuronide	782	–	–
Diosmetin-hexoside	77	–	–
Isorhamnetin-glucosyl-rhamnoside isomer	514	–	–
Isorhamnetin-pentoside	89	–	–
Isorhamnetin-glucosyl-rhamnosyl-glucuronide isomer	185	–	–
Diosmetin-sulphate	243	–	–
Diosmetin	61	–	–
**Total detected compounds**	**4578**	**76**	**–**

n.d., not detectable. The determinations were carried out by UHPLC–MS (ultra-high performance liquid chromatography-mass spectrometry). The results are the average of the concordant values obtained (less than a 5% variation between injections) for two aliquots of each sample injected in triplicate.

**Table 3 foods-09-01533-t003:** Comparison of the productivity and composition of *A. portulacoides* and some typical forage crops.

	Productivity DW (t/ha)	Ash %	Protein %	Lipid %
*Atriplex portulacoides*	10 ^a^	27 ^c^	10–11.2 ^e^	2.1 ^f^
*Medicago sativa* hay	11 ^b^	10.3 ^d^	17 ^d^	2.1 ^g^
Forage grasses (Graminaceae)		9.9 ^b^	8.8 ^b^	1.6 ^b^
Polyphitic meadows		9.7 ^b^	8.9 ^b^	1.6 ^b^
Wheat		7.7 ^b^	8.9 ^b^	2.1 ^b^

^a^ This review (estimation); ^b^ 4 year cycle average [73]; ^c^ refers to shoots in the absence of salinity [25]; ^d^ harvest following the first year [74]; ^e^ refers to leaves [43]; ^f^ refers to leaves [44]; ^g^ refers to leaves [75].

**Table 4 foods-09-01533-t004:** Recommended dietary intake (in g/kg DW) of the minerals that are considered essential for animal nutrition (from CSIRO [77]), and estimations of their intake provided by *A. portulacoides* (stems and/or leaves) and other forage plants.

	Sheep		Cattle		*Atriplex portulacoides*	Corn-Silage	Polyphitic Meadows	Alfalfa
Mineral	Maintenance Requirement (g/kg DW)	Growing, Pregnant, or Lactating Animals (g/kg DW)	Maintenance Requirement (g/kg DW)	Growing, Pregnant, or Lactating Animals (g/kg DW)	(g/kg DW)	(g/kg DW)	(g/kg DW)	(g/kg DW)
Calcium	1.4	7.0	2.0	11.0	7–9 ^a^	2.7 ^h^	6.8 ^f^	15.3 ^f^
Phosphorus	0.9	3.0	1.0	3.8	1.1–1.4 ^a^	2.3 ^h^	2.6 ^f^	2.6 ^f^
Chlorine	0.3	1.0	0.7	2.4	21–31 ^d^	-	-	-
Magnesium	0.9	1.2	1.3	2.2	7–10 ^a^	1.8 ^h^	1.7 ^f^	2.4 ^f^
Potassium	5.0	5.0	5.0	5.0	43–37 ^d^	10.7 ^h^	23.3 ^f^	26.2 ^f^
Sodium	0.7	1.0	0.8	1.2	34–41 ^d^	0.05 ^h^	0.14 ^h^	-
Sulfur	2.0	2.0	1.5	1.5	2 ^e^	1.4 ^h^	2.0 ^h^	-
Cobalt	0.00008	0.00015	0.00007	0.00015	0.008–0.010 ^c^	-	-	-
Copper	0.004	0.014	0.004	0.014	0.012–0.015 ^c^	0.0081 ^h^	0.0129 ^h^	0.011 ^i^
Iodine	0.0005	0.0005	0.0005	0.0005	-	-	-	-
Iron	0.040	0.040	0.040	0.040	0.194–0.715 ^g^	0.200 ^h^	0.184 ^h^	0.382 ^i^
Manganese	0.020	0.025	0.020	0.025	0–0.02 ^a^	0.0381 ^h^	0.0764 ^h^	-
Selenium	0.00005	0.00005	0.00004	0.00004	-	-	-	-
Zinc	0.009	0.020	0.009	0.020	0.045–0.108 ^b^	0.0308 ^h^	0.0276 ^h^	0.029 ^i^

^a^ Ref. [42] leaf content is approximate from the graph; ^b^ Ref. [33], the first value is from stems, and the second value is from leaves; ^c^ Ref. [35], the first value is from leaves, and the second value is from stems; ^d^ Ref. [31], the first value (in the absence of salinity) is from stems, and the second value is from leaves, as approximated from the graph; ^e^ Ref. [78], value from leaves; ^f^ Ref. [73]; ^g^ Ref. [43] the first value is from cultivated plants irrigated with freshwater, and the second value is the average from the natural habitat; ^h^ Ref. [79] the first value is from stems, and the second is from leaves; ^i^ Ref. [80].

**Table 5 foods-09-01533-t005:** Recommended Dietary Allowance (RDA in bold) and Adequate Intake (AI in italics) of minerals for the Italian adult population, and the mineral intake provided by 100 g FW of *A. portulacoides*.

	**Age (Years)**	**Ca (g)**	**P (g)**	**Mg (mg)**	**Na (g)**	**K (g)**	**Cl (g)**	**Fe (mg)**	**Zn (mg)**	**Cu (mg)**	**Se (μg)**	**I (mg)**	**Mn (mg)**	**Mo (μg)**	**Cr (μg)**	**F (mg)**
**Males**	30–59	**1**	**0.7**	**0.24**	*1.5*	*3.9*	*2.3*	**10**	**12**	**0.9**	**55**	*0.15*	*2.7*	*65*	*35*	*3.5*
**Females**	30–59	**1**	**0.7**	**0.24**	*1.5*	*3.9*	*2.3*	**18/10**	**9**	**0.9**	**55**	*0.15*	*2.3*	*65*	*25*	*3.0*
	**FW (g)**	**Ca (g)**	**P (g)**	**Mg (mg)**	**Na (g)**	**K (g)**	**Cl (g)**	**Fe (mg)**	**Zn (mg)**	**Cu (mg)**	**Se (μg)**	**I (mg)**	**Mn (mg)**	**Mo (μg)**	**Cr (μg)**	**F (mg)**
***A. portulac.***	100	0.1	0.016	0.11	0.47	0.5	0.33	5.7	1.0	0.17	-	-	0.13	*-*	*-*	*-*

Values on a daily basis (from http://www.sinu.it, modified). With regard to Fe, in females aged 39–59, the second RDA value refers to postmenopausal women. The assumption of the minerals provided by 100 g FW of *A. portulacoides* were estimated from the averages of the intervals shown in Table 4, assuming a dry weight equal to 12.5% of the fresh biomass: contributions that are ≥ 40% RDA are highlighted in dark grey; those that are ≥ 10% are highlighted in light grey.

**Table 6 foods-09-01533-t006:** Permissible limit of heavy metals and microbiological criteria for edible plants.

**Chemical Hazard**						
**Code**	**Element**	**Limit (mg/kg FW)**		**Law Reference**		
3.1.11	Lead	0.30		Commission Regulation (EC) n. 1881/2006 of 19 December 2006		
3.2.17	Cadmium	0.20		Commission Regulation (EC) n. 629/2008 of 2 July 2008		
**Biological Hazard**						
**Food Category**		**Microorganism**	**Limit (cfu/g)**	**Analytical Reference Method**	**Stage Where the Criterion Applies**	**Law Ref.**
1.2. Ready-to-eat foods able to support the growth of *L. monocytogenes*, other than those intended for infants and for special medical purposes		*Listeria monocytogenes*	*n* = 5 c = 0 absent in 25 g or < 100 cfu/g	EN/ISO 11290-2	Products placed on the market during their shelf-life	(A)
1.19. Pre-cut fruit and vegetables (ready-to-eat)		*Salmonella*	*n* = 5 c = 0 absent in 25 g	EN/ISO 6579	Products placed on the market during their shelf-life	(A)
2.5.1. Pre-cut fruit and vegetables (ready-to-eat)		*E. coli*	*n* = 5 c = 2 m = 100 M = 1000	ISO 16649-1 or 16649-2	Manufacturing process	(A)
1.29. Sprouts		Shiga toxin producing *E. coli* (STEC)	*n* = 5 c = 0 absent in 25 g	CEN/ISO TS 13136	Products placed on the market during their shelf-life	(B)
other spices and herbs		Enterobacter	*n* = 5 c = 1 m = 10 M = 100	ISO 5552	Import or production/packaging or wholesale/establishment/retail	(C)
other spices and herbs		*Clostridium perfrigens*	*n* = 5 c = 1 m = 100 M = 1000	ISO 7937	Import or production/packaging or wholesale/establishment/retail	(C)
other spices and herbs		*Bacillus cereus*	*n* = 5 c = 1 m = 1000 M = 10,000	ISO 7932	Import or production/packaging or wholesale/establishment/retail	(C)

*n*, samples required; c, samples admitted with acceptable counts; m, lower limit; M, maximum limit. Criteria: satisfactory, if all of the values observed are < m; acceptable, if a maximum of c/n values are between m and M, and the rest of the values observed are ≤ m; unsatisfactory, if one or more of the values observed are > M, or more than c/n values are between m and M. (A) Commission Regulation (EC) n. 2073/2005 of 15 November 2005; (B) Commission Regulation (EC) n. 209/2013 of 11 March 2013; (C) Commission Recommendation of 19 Dec 2003 (2004/24/EC).

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
