# Peer review of "Functional Food from Endangered Ecosystems: Atriplex portulacoides as a Case Study"

_foods, 2020, doi:10.3390/foods9111533_

Round 1
Reviewer 1 Report
The manuscript entitled: Functional food from endangered ecosystems: Atriplex portulacoides as case study
Overall comments: The manuscript touches very interesting and perspectives problem of new utility of Atriplex portulacoides to production of functional food. Unfortunately, the reviewed manuscript contain some information that are completely unnecessary. Reading the title of the manuscript, one would expect the authors to focus on describing the functional ingredients contained in the plant, and much of the information relates to the description of the typical nutrients contained in the plant. It should be clearly stated in the introduction how the authors understand functional food. In my opinion, the authors described completely unnecessary properties of the plant as source of animal feed.
At the end of the introduction section, I suggest that the authors clearly emphasize as the aim of the research the presentation of the functional properties of A. portulacoides. Also in the summary section, I suggest clearly describing and highlighting the functional properties of A.P.
Detail comments:
Line 167: table 1 entitled “Dry biomass and approximate composition….” . The protein content shown in the table was estimated from the measurement of the N content, and this result was then calculated as an approximation to the amount of protein contained in the plant. Are the results of measuring the protein content in the plant available in the literature, determined by a method other than determination of nitrogen content?
I cannot find information about the starch content in Atriplex portulacoides . Is there starch presented in the plant?
In this study l cannot find the information on the digestibility of nutrients by the human digestive system. What is the vitamin content in individual parts of the plant?
The manuscript requires a minor revisions in accordance with my comments.
Author Response
Point-by-point reply to reviewer’s comments for the revision of the manuscript entitled: “Role of carboxylic group pattern on protein surface in the recognition of iron oxide nanoparticles: a key for protein corona formation” (ID: foods-952060)
Reviewer 1:
Overall comments: The manuscript touches very interesting and perspectives problem of new utility of Atriplex portulacoides to production of functional food. Unfortunately, the reviewed manuscript contains some information that are completely unnecessary. Reading the title of the manuscript, one would expect the authors to focus on describing the functional ingredients contained in the plant, and much of the information relates to the description of the typical nutrients contained in the plant. It should be clearly stated in the introduction how the authors understand functional food. In my opinion, the authors described completely unnecessary properties of the plant as source of animal feed.
At the end of the introduction section, I suggest that the authors clearly emphasize as the aim of the research the presentation of the functional properties of A. portulacoides. Also in the summary section, I suggest clearly describing and highlighting the functional properties of A.P.
We thank the reviewer for his appreciations and critical comments. In the revised version of the manuscript, according to reviewer suggestion, we emphasized our aims. As A. portulacoides is not available as common food, the functional properties of the plant need to be supported by other traits, without which it cannot be exploited in a realistic application as food. Cultivability and toxicological safety are indispensable factors for promoting the approval of its use at industrial scale. Notably, the historical use of the plant is a mandatory issue required by EFSA for authorizing its commercial exploitation as novel foods. In should be noted that documented evidences of the use of this plant are not easily available and are dispersed in literature focused on other plants.
However, functional properties of A. portulacoides were also reported in detail within the limit of the present knowledge (please, see par. 6.2), which need to be improved, as remarked in the conclusion of the revised version of the manuscript.
The introduction of details on the effects of functional compound extracted from the plant on human biochemical pathways are at the moment unavailable. Authors hope that the proposed work paves the way for the development of further studies on the impact of A. portulacoides as functional food.
Detailed comments:
Point 1. Line 167: table 1 entitled “Dry biomass and approximate composition….” . The protein content shown in the table was estimated from the measurement of the N content, and this result was then calculated as an approximation to the amount of protein contained in the plant. Are the results of measuring the protein content in the plant available in the literature, determined by a method other than determination of nitrogen content?
Answer 1. Authors thank the reviewer for the comment. Table 1 was rearranged in the revised version of the manuscript and a note (please, see lines 204-206) was introduced to clarify that some values of protein content were not reported in the original reference (please, see ref. [40, 42]). Another reference (please, see ref. [33, 43]), provided the final data from which the protein content was estimated.
Point 2. I cannot find information about the starch content in Atriplex portulacoides . Is there starch presented in the plant?
Answer 2. Unfortunately, no data are available in literature on this aspect. Reboredo et al. (2012) studied the starch granules in the chlorenchyma of chloroplasts of A. portulacoides with respect to the effect of Zn treatments, but no quantification is available. In our opinion, starch could be present in fruits, that are small but numerous, even if, at the best of author knowledge, no study exists on the composition of fruits of A. portulacoides.
Point 3. In this study l cannot find the information on the digestibility of nutrients by the human digestive system. What is the vitamin content in individual parts of the plant?
Answer 3. Unfortunately, no data are available in literature on this aspect of A. portulacoides. We remarked these issues at lines 559-565 for the revised manuscript.
Point 4. The manuscript requires a minor revisions in accordance with my comments.
Answer 4. Authors thank the reviewer.
Reviewer 2 Report
The submitted manuscript deals with an important issue that is potential use of high-salinity- and water deficit-tolerant plants as food and fodder source.
The review, based solely on literature data, is well-written, honestly judging risk/benefit ratio in reference to current EU regulations. Cultivation prospects of Atriplex portulacoides as a crop plant have been also discussed.
The paper is worth publishing, though some questions should be addressed before the final version is issued.
Section 6.1.
As leaves constitute c. 13.7% of the crop (Table 1); is it rational to compare the content of the protein and lipid fractions in leaves of A. portulacoides to those in the other crop plants (Table 3)?
Section 6.2.
Please, be more critical with literature data. In fact, references 89 and 90 did not support the claim that hopenone could be taken into consideration as anticancer agent.
Inhibition of cholinesterase by 20-hydroxyecdysone, described by Nejma et al. (2015), was very modest one, as well as antibacterial activity of the compound.
Section 7., line 429 – please remove “]”.
References need corrections, e.g.:
line 635 – last page missing – “169.” -> 169-173.
line 763 – pagination incomplete
line 788 – erroneous citation – article number missing
Author Response
Point-by-point reply to reviewer’s comments for the revision of the manuscript entitled: “Role of carboxylic group pattern on protein surface in the recognition of iron oxide nanoparticles: a key for protein corona formation” (ID: foods-952060)
Reviewer 2
The submitted manuscript deals with an important issue that is potential use of high-salinity- and water deficit-tolerant plants as food and fodder source. The review, based solely on literature data, is well-written, honestly judging risk/benefit ratio in reference to current EU regulations. Cultivation prospects of Atriplex portulacoides as a crop plant have been also discussed. The paper is worth publishing, though some questions should be addressed before the final version is issued.
Section 6.1.
Point 1. As leaves constitute c. 13.7% of the crop (Table 1); is it rational to compare the content of the protein and lipid fractions in leaves of A. portulacoides to those in the other crop plants (Table 3)?
Answer 1. Authors thank the reviewer for the appropriate comment. As shown in Fig. 1, A. portulacoides develops subreptant branches (which become quite robust and strongly lignified), and ending shoots, rich in leaves and with green/red tender branches. We considered that the composition of leaves better approximate the composition of a potential harvest, which should be composed by leaves and tender branches. Indeed, the upper part of the canopy should be harvested for the use as food or feed, since humans and animals do not accept the ligneous part. Thus, even if tender tissues are a minor part of the whole dry biomass due to their high water content, they compose most of the harvest. Therefore, authors suggest that leaf composition represents the best approximation for the proposed use of A. portulacoides as potential crop. The estimated productivity observed in nature provided an indication of the potential capacity of this perennial plant to re-generate new shoots after harvesting. The new harvest should be performed before the lignification of new shoots. The practical comment was introduced in the revised manuscript (please, see lines 570-574).
Point 2. Section 6.2. Please, be more critical with literature data. In fact, references 89 and 90 did not support the claim that hopenone could be taken into consideration as anticancer agent.
Inhibition of cholinesterase by 20-hydroxyecdysone, described by Nejma et al. (2015), was very modest one, as well as antibacterial activity of the compound.
Answer 2. According to authors opinion, biologically active substances of functional foods are not expected to act as drugs for treating a disorder or pathology, but they can contribute to prevent health problems on long term and constant assumption, possibly acting as a synergistic combination of mild effects. Anyway, the statements regarding hopenone and 20-hydroxyecdysone were mitigated in the revised text (please, see lines 419-423). Consistently, the results by Nejma et al. were considered acceptable, and worth to be mentioned.
Point 3. Section 7, line 429 – please remove “]”.
Answer 3. The text was modified as suggested (please, see line 493 of the revised manuscript).
Point 4. References need corrections, e.g.: line 635 – last page missing – “169.” -> 169-173.
Answer 4. The text was modified as suggested (please, see line 723 of the revised manuscript).
Point 5. line 763 – pagination incomplete
Answer 5. The text was modified as suggested (please, see line 848 of the revised manuscript).
Point 6. line 788 – erroneous citation – article number missing
Answer 6. The text was modified as suggested (please, see line 874 of the revised manuscript).
Reviewer 3 Report
The present manuscript aims to review thoroughly the composition and productivity of A. portulacoides, in order to point out the potential exploitation as functional food. The objective is clear and the authors develop the story in a very interesting and flawless way. However some aspects need to be reviewed before its publication.
Please find below a detailed list for revision:
Line 30: Keep consistency when numbering the section titles
Line 80: I would suggest to name the province of the Netherlands referred in the cited article as North Holland.
Section 3: When reading, it seems that purslane was only a common herb for the Netherlands and Italy. However, it was used more widely. For example by ancient Greeks, or even Aboriginal Australians. With this section, the authors intend to point out that this would not be a newly introduced crop culturally or ecologically. Therefore it is necessary to mention that is has been used so widely in the past. Obviously, there is no need to specify every single use of purslane throughout the history.
Table 1. Keep consistency in the position of the units (I suggest in the first column). Align the columns and rearrange all values in order to make the table easy to read. The sub index “a” is not present in the table. Please delete the sub index explanation or place the sub index in the correspondent place.
Table 4. Keep consistency using brackets in the unit specification. Remove sub index h in the Corn-silage cell and keep the sub index only in the cells containing values. Indicate empty cells with abbreviations (for example “n.s.” non studied, “u” unknown) or symbols (for example “-”). Keep this consistent in other tables if necessary
Section 6.2 Human Nutrition: In this section the individual components of purslane have been widely cross-referenced with beneficial effects. However, the manuscript does not show enough direct correlation of purslane consumption or use and health improvement. Only two in vitro studies have been mentioned.
There are clinical studies that directly correlate the purslane (extract or seeds) consumption with type-II diabetes improvement. The biochemical mechanisms have also been assessed in vitro. There are also in vivo studies that show for example that evaluate the antioxidant and immuno-enhancing activities of purslane polysaccharides in gastric cancer rats. The antioxidant and antiproliferative activities of Purslane seed oil have been also recently studied in vitro. This section needs to be revisited.
Is there any information about the absorption and metabolism of purslane by the human gastrointestinal tract?
Table 5: Values in mg that contain 3 or more units (Ca, P, and Mg) can be expressed in g like Na, K, and Cl. This is also applicable for Iodine. Minerals intake (units? Is it mg?) provided by 100 g FW of A. portulacoides
Table 6: All the references in the table are not listed accordingly. Please complete the reference list following the journal guidelines.
Line 429: Delete brackets
Line 460: use italics for in vivo (keep this consistent throughout the manuscript)
Line 487: "Authors declare no conflict of interest" is an example of how to make the declaration of interest statement correctly.
Author Response
Point-by-point reply to reviewer’s comments for the revision of the manuscript entitled: “Role of carboxylic group pattern on protein surface in the recognition of iron oxide nanoparticles: a key for protein corona formation” (ID: foods-952060)
Reviewer 3
The present manuscript aims to review thoroughly the composition and productivity of A. portulacoides, in order to point out the potential exploitation as functional food. The objective is clear and the authors develop the story in a very interesting and flawless way. However some aspects need to be reviewed before its publication,
Please find below a detailed list for revision:
Point 1. Line 30: Keep consistency when numbering the section titles
Answer 1. Authors thank the reviewer for the suggestion. The text was modified as suggested (please, see line 30 of the revised manuscript).
Point 2. Line 80: I would suggest to name the province of the Netherlands referred in the cited article as North Holland.
Answer 2. The text was modified as suggested (please, see line 99 of the revised manuscript).
Point 3. Section 3: When reading, it seems that purslane was only a common herb for the Netherlands and Italy. However, it was used more widely. For example by ancient Greeks, or even Aboriginal Australians. With this section, the authors intend to point out that this would not be a newly introduced crop culturally or ecologically. Therefore it is necessary to mention that is has been used so widely in the past. Obviously, there is no need to specify every single use of purslane throughout the history.
Answer 3. Authors thank the reviewer. In order to support the acceptance as a novel food, the historical use has to be as robust and documented as possible. Mentioned cases referred to the use this herb in recent times. Even if several websites deals with medicinal and edible plants and generically report on the food use of A. portulacoides, no production of this plant is reported, and no restaurant offers preparations with this ingredient. In fact, the use of A. portulacoides is a local and forgotten tradition, which requires a new regulatory validation to be reintroduced as widespread and commercially exploitable food practice.
Point 4. Table 1. Keep consistency in the position of the units (I suggest in the first column). Align the columns and rearrange all values in order to make the table easy to read. The sub index “a” is not present in the table. Please delete the sub index explanation or place the sub index in the correspondent place.
Answer 4. Authors thank the reviewer for the suggestion. The revision of This table led authors to introduce some changes that we hope make it more readable, but also more carefully compiled (please, see the revised version of the manuscript). Doubtful data have been revised and a new reference was added. We do hope that this version finds the reviewer approval.
Point 5. Table 4. Keep consistency using brackets in the unit specification. Remove sub index h in the Corn-silage cell and keep the sub index only in the cells containing values. Indicate empty cells with abbreviations (for example “n.s.” non studied, “u” unknown) or symbols (for example “-”). Keep this consistent in other tables if necessary
Answer 5. Authors thank the reviewer for the suggestion. The suggested modifications have been adopted in the revised manuscript (please, see Table 4 of the revised manuscript).
Point 6. Section 6.2 Human Nutrition: In this section the individual components of purslane have been widely cross-referenced with beneficial effects. However, the manuscript does not show enough direct correlation of purslane consumption or use and health improvement. Only two in vitro studies have been mentioned.
There are clinical studies that directly correlate the purslane (extract or seeds) consumption with type-II diabetes improvement. The biochemical mechanisms have also been assessed in vitro. There are also in vivo studies that show for example that evaluate the antioxidant and immuno-enhancing activities of purslane polysaccharides in gastric cancer rats. The antioxidant and antiproliferative activities of Purslane seed oil have been also recently studied in vitro. This section needs to be revisited.
Answer 6. Authors thank the reviewer for the comment. We did not find any reference consistent with reviewer’s comment. Possibly, the reviewer considered other papers related with common purslane (Portulaca oleracea) that belongs to another plant family and occurs in a completely different habitat. In order to make clearer this distinction, often problematic, authors introduced a specific comment (please, see lines 86-94 of the revised manuscript). However, a further control of literature allowed us to add an interesting contribution concerning phenolic compounds and their activity on some enzymes (please, see lines 274-280 and 448-450).
Point 7. Is there any information about the absorption and metabolism of purslane by the human gastrointestinal tract?
Answer 7. Unfortunately, information about absorption and metabolism is not yet available.
Point 8. Table 5: Values in mg that contain 3 or more units (Ca, P, and Mg) can be expressed in g like Na, K, and Cl. This is also applicable for Iodine. Minerals intake (units? Is it mg?) provided by 100 g FW of A. portulacoides
Answer 8. Authors thank the reviewer for the suggestion. The Table was rearranged accordingly.
Point 9. Table 6: All the references in the table are not listed accordingly. Please complete the reference list following the journal guidelines.
Answer 9. Authors thank the reviewer for the comment. Anyway, authors think that is convenient for the reader to find references in notes of the Table and cited regulations are usually reported by authority, number of regulation and year of publication. Author guidelines of the journal do not give specific indications of these special references, which are not scientific authored sources. However, we will comply the editor decision on this point.
Point 10. Line 429: Delete brackets
Answer 10. The text was modified as suggested (please, see line 493 of the revised manuscript).
Point 11. Line 460: use italics for in vivo (keep this consistent throughout the manuscript)
Answer 11. The text was modified as suggested (please, see line 524 of the revised manuscript and also 565-569).
Point 12. Line 487: "Authors declare no conflict of interest" is an example of how to make the declaration of interest statement correctly.
Answer 12. The text was modified as suggested (please, see line 579 of the revised manuscript).
Round 2
Reviewer 3 Report
I would like to thank the Authors for all the clarifications. Manuscript has been improved accordingly and is ready for publication. Congratulations.